# Assessment of Surface Roughness, Color, and Bonding Efficacy: Self-Adhesive vs. Conventional Flowable Resin

**DOI:** 10.3390/polym16182556

**Published:** 2024-09-10

**Authors:** Caroline de Farias Charamba Leal, Beatriz Barros Viana, Samille Biasi Miranda, Renally Bezerra Wanderley e Lima, Cleyton Cézar Souto Silva, Rodrigo Barros Esteves Lins, André Ulisses Dantas Batista, Ana Karina Maciel de Andrade, Marcos Antônio Japiassú Resende Montes

**Affiliations:** 1Departament of Dental Materials, Faculty of Dentistry, University of Pernambuco, Recife 50100-130, PE, Brazil; caroline.charamba@upe.br (C.d.F.C.L.); samille.biasi@upe.br (S.B.M.); 2Departament of Restorative Dentistry, Federal University of Paraíba, João Pessoa 58051-900, PB, Brazil; beatriz.barros@academico.ufpb.br (B.B.V.); renallywanderley@gmail.com (R.B.W.e.L.); andreulisses@yahoo.com.br (A.U.D.B.); kamandrade@hotmail.com (A.K.M.d.A.); 3Department of Clinical Nursing, Federal University of Paraíba, João Pessoa 58051-900, PB, Brazil; cleyton.souto@academico.ufpb.br; 4Department of Restorative Dentistry, School of Dentistry, Federal University of Alagoas, Maceió 57072-900, AL, Brazil; rodrigo.lins@foufal.ufal.br

**Keywords:** self-adhesive flowable composite, toothbrushing, microtensile bond strength, surface properties, color

## Abstract

This in vitro study aimed to analyze the surface roughness (Ra) and color stability (ΔEab, ΔE00) following simulated mechanical brushing and to evaluate the microtensile (μTBS) of self-adhering resin flowable (SARF) to dentin. The selected materials were Constic, Yflow AS, and Tetric N flow (TNF/control). Thirty composite resin cylinders were fabricated for surface property evaluation. Ra and color were assessed both before and after simulated brushing. The thresholds of 50:50% perceptibility and acceptability of color differences in the evaluated resins were assessed. For μTBS analysis, fifteen molars were selected, sectioned to expose flat dentin surfaces, and restored according to the manufacturers’ instructions for microtensile testing. There were statistically significant differences in Ra among the groups, with Constic exhibiting the highest Ra value (0.702 µm; *p* < 0.05), whereas Yflow AS presented the lowest Ra value (0.184 µm). No statistically significant difference in color was observed among the groups (*p* > 0.05). The 50:50% perceptibility and acceptability thresholds were set at 1.2 and 2.7 for ΔEab and 0.8 and 1.8 for ΔE 00. All the results fell within the acceptable limits. The mean μTBS values of Constic, Yflow AS, and TNF were 10.649 MPa, 8.170 MPa, and 33.669 MPa, respectively. This study revealed increased Ra and comparable color stability among all the tested composite resins after abrasion. However, the SARF exhibited lower μTBS compared to conventional using an adhesive system.

## 1. Introduction

Resin composites have undergone significant advancements since their inception, evolving to meet the demands of modern dentistry by enhancing their physical and optical properties. These materials have become the preferred choice for aesthetic restorations and areas subjected to significant chewing stresses, offering a versatile solution for replacing lost dental tissue resulting from caries and restoring long-term harmony, function, and stability of the dentition [1,2,3]. Despite their widespread use, composite resin restorations are susceptible to various types of failure, including restoration fracture, discoloration, adhesive failure to the tooth, and secondary caries [4,5]. Many of these failures are correlated with the sensitivity of the adhesive technique to moisture [6]. In response to these challenges, self-adhesive flowable resins (SAFRs) have emerged as promising alternatives aimed at simplifying restoration techniques [7].

SAFRs have similar indications to conventional flowable resins (CFRs), such as class I, class III, and class V restorations, sealants, and repairs [7,8,9,10]. The self-adhesiveness of this new class of composite resin is conferred by the addition of glycerol phosphate molecules, which condition the dental structure while also having an affinity for the dental element calcium and methacrylate groups that copolymerize with other methacrylate monomers [11]. However, it is important to note that tissue demineralization is partial, meaning that there is no complete removal of the smear layer, incorporating it into the adhesive interface, unlike the usually applied technique of total acid conditioning, where the smear layer is entirely removed [12]. The success of restorative treatment depends on the interaction between the restorative material and dental tissue, which is commonly evaluated through bond strength tests [10].

Compared with traditional bonding techniques, the microtensile μTBS (μTBS) of SAFR remains a subject of investigation, with conflicting findings [13,14,15]. Brueckner et al. [13] conducted a dentin and enamel bonding strength test after simulated aging with thermocycling, comparing SAFR with a conventional flowable resin used with an adhesive system. The SAFRs evaluated (vertise flow, fusion liquid dentin, and an experimental SAFR) exhibited lower adhesion performance to dental tissues than conventional flowable resins in Class V cavities. In turn, Abdelraouf, Mohammed, and Abdelgawad [14], in a similar study in which the testing area was prepared after the vestibular face of the included molars was cut, also reported that the evaluated SAFR (Dyad flow) had inferior bonding strength to dental substrates compared with conventional flowable resins. Hayashi et al. [10] reported that the adhesion values of SARFs are greater when prior acid etching is performed, according to an in vitro study conducted on bovine teeth. This study highlights the need for further investigations into specific resin formulations and insertion techniques to assess their clinical viability. For example, it was found that inserting one SARF (fusion liquid dentin) was easier than inserting another (vertise flow) [12]. Elraggal et al. [16] conducted shear and microtensile bond strength tests to evaluate the adhesion of SARFs to dentin after thermocycling. The resins tested (Surefil One and Vertise Flow) demonstrated inferior results compared to conventional flowable resin. Studies evaluating the μTBS of restorative materials are crucial for subsequent clinical research, as stated by Brueckner et al. [13] and Elraggal et al. [16]. In vitro studies reflect the clinical performance of resins, and we can only use them in vivo after promising laboratory results are obtained.

The wear of dental restoration can be influenced by factors such as occlusal contact, food bolus friction, and tooth brushing [17]. The abrasive challenge generated by tooth brushing can increase the roughness (Ra) of composite resins. Consequently, the literature highlights that simulated abrasion is a well-established model capable of simulating clinical conditions [18]. SARFs demonstrated high luster and a smooth surface after polishing [19]. However, after 60 min of simulated brushing with an electric toothbrush and Colgate Total toothpaste, these characteristics were altered. The evaluated SARF exhibited a Ra value above 0.2 µm, leading to surface irregularities, filler dislodgement, and protrusion [19]. Malavasi et al. [18] conducted simulated brushing, where the samples were brushed for 20,000 cycles using a soft-bristled toothbrush and Colgate Total toothpaste. After the abrasive challenge, the surface roughness values of the SARFs were greater (0, 24, and 0.15 µm) than those of the conventional resin (0.13 µm) used as a control. This increase in roughness was attributed to the smaller filler particles in the conventional resin, which provided better support and protection for the fillers. As the abrasion process removes the organic matrix of the resins, the larger filler particles in the SARFs are more exposed, leading to greater surface irregularities [18]. The increased roughness of restorative materials increases the susceptibility of the surface to plaque accumulation, staining, and deterioration of the aesthetic properties of the material [20]. Color represents an important aspect of the aesthetic success of restorations. A previous study [19] highlighted that simulated brushing affected the color stability of self-adhesive flowable resins (SARFs), with the SARFs showing the highest ΔE values after the abrasive challenge. This increase in ΔE was attributed to the abrasivity process itself; during brushing, the samples were immersed in a slurry (toothpaste and distilled water), which can lead to water absorption and potentially impact color stability [19].

A previous study suggested further exploration of the physical and mechanical properties of SARFs to gain a deeper understanding of their polymerization process [21]. In vitro studies are crucial for the development of new materials, such as self-adhesive flowable resins (SAFRs), as they can provide essential information for further testing and ensure safety in future clinical trials [22]. This study provides insights into the performance of SARFs by comprehensively evaluating their surface roughness (Ra), color stability, and microtensile bond strength (μTBS) in comparison to CFRs. Through these evaluations, the results will either reaffirm or challenge the use of self-adhesive resins, offering a clearer understanding of their suitability and effectiveness in clinical practice. While previous research has focused on specific aspects of SARFs, such as bond strength or color stability, this study adopts a more holistic approach by simultaneously assessing these critical factors. The use of both the CIELAB and CIEDE2000 methods for color assessment contributes to a more nuanced understanding of color stability across various SARFs and CFRs. Furthermore, this study addresses the mechanical performance of SARFs in terms of μTBS, highlighting a gap in the literature, as no studies evaluating the μTBS of Yflow SA (a specific SARF) were found. The findings of this study can provide scientific evidence on the performance of SARFs, contributing to a better understanding of their characteristics and limitations. Thus, this study aimed to identify the behavior of SAFR in terms of Ra, color stability after simulated brushing, and μTBS to dentin. The null hypotheses are as follows: (a) there is no significant difference in the Ra of the composite resins after simulated brushing, (b) there is no significant difference in the color of the composite resins after simulated brushing, and (c) there is no significant difference in the μTBS between the SAFR and CFR.

## 2. Materials and Methods

An in vitro study was conducted to assess the Ra, color, and μTBS to dentin of the SAFRs Constic (DMG, Hamburg, Germany; Lot No. 232801) and Yflow SA (Yller Biomateriais, Pelotas, RS, Brazil; Lot. No. 00009968) in comparison with the CFR Tetric N-flow (TNF) (Ivoclar Vivadent, Barueri, SP, Brazil; Lot No. Z020DP, Z019WR). The Constic consists of barium glass in a matrix of dental resins based on Bis-GMA, EBADMA, UDMA, HEMA, TEGDMA, HDMA, and MDP, along with pigments, additives, and a catalyst (DMG). Yflow contains methacrylate monomers, acid monomers, inorganic fillers, pigments, initiators, and stabilizers (Yller biomaterials, Pelotas, Brazil), and TNF contains dimethacrylates (including TEGDMA), barium oxide, ytterbium trifluoride, highly dispersed silica, mixed oxides, pigments, catalysts, and stabilizers (Ivoclar Vivadent, Bunkyo, Japan). This study was carried out in two stages; initially, tests for Ra and color analysis were performed before and after simulated brushing. A μTBS test was subsequently conducted to evaluate the bonding resistance of the investigated resins to dentin (CAAE: 48791621.1.0000.5188).

### 2.1. Initial Analysis of Surface Roughness and Color

The analyses of Ra and color stability were conducted with a sample size of 10 samples per group. The number of samples was determined on the basis of a previous study [19] that evaluated the roughness and color stability of the SARF resin compared with the CFR. This study adopted a power of 80% and a significance level of 0.05, with minimum differences of 0.04 in roughness and 0.96 in color. Although the required number of samples was 5, a sample size of 10 samples per group was adopted to achieve greater precision in accordance with the recommendations outlined in the RoB assessment tool for laboratory studies on dental materials [22]. The samples were divided according to the type of restorative composite resin into three groups, each containing 10 samples: Group 1A (TNF), Group 2A (Constic), and Group 3A (Yflow SA).

The samples were prepared with a cylindrical Teflon matrix measuring approximately 2 mm thick and 6 mm in diameter. The matrix was filled with the restorative material in a single increment. After material insertion, a polyester strip was pressed onto the surface using a glass plate to ensure that the surface was flat and free from bubbles. Polymerization was carried out using Emitter C equipment (SCHUSTER, Santa Maria, RS, Brazil), with the light intensity measured above 800 mW/cm^2^ using an RD-7 radiometer (ECEL, Ribeirão Preto, SP, Brazil). The samples were subsequently subjected to finishing and polishing using the Sof-Lex system (3 M, Oral Care, St. Paul, MN, USA) and felt discs (FGM, Joinville, SC, Brazil), followed by immersion in distilled water for 24 h.

The initial analysis of Ra of the samples was performed using a profilometer (Surftest SJ-301, Mitutoyo, Japan). The samples were individually placed on a glass plate with utility wax using a press with a load of 0.5 kgf for 3 s. Three random readings per sample were taken (horizontal, transverse, and vertical), with the sample rotated during each reading. The readings were taken through the center of the surface of each sample. The recorded Ra value was the average of the readings. Tooth color was assessed using a spectrophotometer (Easyshade Advance 4.0, VITA Zahnfabrick, Bad Säckingen, Germany) against a black background. Calibration of the device was conducted after every three measurements. The CIELab color coordinates, including L*, a*, and b*, as defined by the International Commission on Illumination (CIE), were recorded. These coordinates are extensively utilized in the dental literature. The L* value signifies the psychometric brightness ranging from black to white (achromatic coordinate), whereas the a* (green–red coordinate) and b* (blue–yellow coordinate) values represent the psychometric chroma coordinates, indicating hue and chroma, respectively [23]. Measurements were conducted three times for each sample, and the resulting data were averaged. To enhance correlation with visual perception, the International Standard Organization (ISO) and the International Commission on Illumination (CIE) jointly recommend employing the CIEDE2000 color difference formula for determining the total color difference, which is based on the CIELAB color space [24]. Recent studies have further corroborated that the CIEDE 2000 formula better represents color differences perceived by the human eye than the CIELAB [24] formula. Thus, in this study, color differences were assessed using two parameters: CIEDE 2000 (ΔE 00) and CIELAB (ΔE ab). The CIELAB color difference (ΔE ab) was calculated using the following equation:$$∆E={\left[{\left(∆L\right)}^{2}+{\left(∆a\right)}^{2}+{\left(∆b\right)}^{2}\right]}^{\frac{1}{2}}$$

The CIEDE2000 color difference (∆E 00) was computed using an Excel spreadsheet implementation of the CIEDE2000 color difference formula developed by Sharma [25]. ΔE00 (CIEDE 2000) was calculated using the following equation:$${\Delta E}_{00}={\left[{\left(\frac{\Delta L^{\prime }}{K_{L}S_{L}}\right)}^{2}+{\left(\frac{\Delta C^{\prime }}{K_{C}S_{C}}\right)}^{2}+{\left(\frac{\Delta H^{\prime }}{K_{H}S_{H}}\right)}^{2}+RT\left(\frac{\Delta C^{\prime }}{K_{C}S_{C}}\right)\left(\frac{\Delta H^{\prime }}{K_{H}S_{H}}\right)\right]}^{1/2}$$

ΔL’, ΔC’, and ΔH’ represent variations in brightness, color intensity, and hue among the samples being compared. RT serves as a rotation function for elucidating how chroma and hue differences interact, particularly within the blue spectrum. S L, S C, and SH denote weighting functions, whereas K L, K C, and K H refer to terms that need an adjustment on the basis of experimental conditions [26]. The parametric factors kL, kC, and kH function as correction terms for experimental variations and will all be standardized to 1.0 under reference conditions as determined by the CIE technical report.

The color adaptability was assessed on the basis of the color disparities observed between the teeth and the composite materials, considering thresholds of 50%:50% perceptibility and 50%:50% acceptability. Under these thresholds, the 50%:50% color difference (PT) perceptibility threshold denotes the degree of color variation that can be perceived by 50% of observers in controlled conditions, whereas the remaining 50% of observers perceive no discernible difference in color between the compared objects. Similarly, the 50:50% color difference (AT) acceptability threshold signifies the level of color difference deemed acceptable by 50% of observers under controlled conditions, with the remaining 50% of observers opting to either replace or correct the restoration [26]. The ΔEab—CIELAB values were assessed on the basis of 50%:50% perceptibility and 50%:50% acceptability criteria, with thresholds set at 1.2 and 2.7 according to the specifications outlined in ISO/TR 28642:2016. Additionally, the ΔE 00-CIEDE 2000 values were examined using the same perceptibility and acceptability standards, with corresponding thresholds established at 0.8 and 1.8, respectively, as indicated by prior research [26].

### 2.2. Simulated Toothbrushing

Simulated toothbrushing was conducted using a specific device (Biopdi, São Carlos, SP, Brazil) capable of performing back-and-forth movements across ten articulated arms using pulleys with attached dental brushes. Each complete movement was considered one cycle. The toothpaste used was Colgate Cavity Protection (Colgate-Palmolive Company, New York, NY, USA), which includes 1500 ppm fluoride, calcium carbonate, sodium lauryl sulfate, sodium saccharin, tetrasodium pyrophosphate, sodium silicate, polyethylene glycol, sorbitol, carboxymethyl cellulose, methylparaben, propylparaben, aromatic composition, and water. It also contains sodium monofluorophosphate (MFP^®^). The toothpaste was applied as a suspension and diluted in distilled water at a ratio of 2:1 using a mechanical agitator. The samples were subjected to 20,000 cycles of brushing. The product was paired with Oral-B Indicator 40 Soft toothbrushes (Gillette do Brasil Ltd.a., São Paulo, SP, Brazil), featuring nylon bristles with handles cut at the midpoint to fit into the machine. Following the brushing cycles, the samples were removed from the machine and rinsed under running water. Given that individuals brush their teeth three times a day, a total of 5465 brushing cycles occur over a year. Therefore, the chosen brushing period in this study is equivalent to approximately 3 years and 6 months in vivo [27].

### 2.3. Roughness and Color Analysis after Simulated Brushing

The Ra and color measurements were conducted following the same criteria as those used for the initial assessment. The mean Ra values were recorded, and color differences between the coordinates were calculated using a specific formula.

### 2.4. Microtensile Test

For the μTBS evaluation, 15 human molars extracted upon the recommendation of a dentist were obtained following the guidelines of Resolution 466/12 of the National Health Council/MS after approval from the Research Ethics Committee—CCS/UFPB. The molars were divided into three groups according to the restorative material used, namely, Group 1B (TNF), Group 2B (Constic), and Group 3B (Yflow SA). The selected teeth were cleaned and embedded in acrylic resin cylinders to facilitate handling during preparation. The occlusal enamel was removed using a diamond disc (Extec, Enfield, CT, USA) mounted on a precision cutter (Labcut 1010, Extec Corp., Enfield, CT, USA) under water cooling to expose the dentin surface, resulting in a flat dentin surface. The samples were subsequently ground with 600-grit silicon carbide paper (Saint-Gobain Abrasivos, Guarulhos, SP, Brazil) on a water-cooled polisher (ERIOS Polisher-27000, ERIOS, São Paulo, SP, Brazil) for 60 s to provide a homogeneous layer of dentin slurry [28].

The adhesive system Adper Single Bond Plus (3 M ESPE) was applied following the manufacturer’s recommended technique, only with TNF resin. A metal matrix band was placed, and each sample received two increments, each 2.0 mm thick, of resin. Photoactivation was performed using an Optlux Plus light-curing unit (GNATUS) with a light intensity of 800 mW/cm^2^, as determined using a Demetron radiometer (Model 100, Demetron, Brea, CA, USA), resulting in the formation of a block approximately 4.0 mm in height. The restorative technique for each composite resin was performed according to the manufacturer’s instructions. For the TNF, the procedure involves applying an adhesive to the enamel and dentin and then placing the resin into the cavity using the incremental technique. Each increment should be between 1.5 and 2 mm thick, with a curing time of 20 s per increment using halogen light with a minimum intensity of 500 mW/cm^2^. For Constic, the material is applied to the cavity surface by pressing the syringe and rubbing the resin for 25 s to form a thin layer (0.5 mm). Any excess material was removed if necessary, followed by light curing for 20 s. Additional increments of up to 2 mm are then applied, and the samples are cured for 20 s each. For Yflow, after dental cleaning, the product is directly applied into the cavity using a syringe with an applicator tip. The resin is placed in increments of up to 2 mm, with each increment being light-cured for 40 s.

With a diamond disc (Extec, Enfield, CT, USA) mounted on a precision cutter (Labcut 1010, Extec, Enfield, CT, USA) under water cooling, the samples were longitudinally cut in the mesio–distal and vestibulo–lingual directions to obtain prism-shaped samples measuring approximately 1 mm wide, 1 mm deep, and 10 mm high. These specimens consisted of two arms, one composed of the restorative material and the other composed of the dentin substrate, bonded together by an adhesive interface [28]. Seven sticks were obtained from each tooth, totaling 35 in the group where TNF resin was used. The sample size calculation was based on a previous study [29], adopting a power of 80% and a significance level of 0.05, resulting in 35 samples, which were able to detect a difference of up to 5.3. The groups in which self-adhesive resins were used encountered problems in obtaining the required number of sticks, which will be discussed further.

Using a cyanoacrylate-based adhesive (Super Bond Gel–Loctite, Itapevi, Brazil Ltd.) and an accelerator substance, the ends of each sample were fixed to the grips of the microtensile testing device, leaving the bond interface free. The grips attached to the sample were subsequently positioned in a testing machine using a 500 N load cell, which was activated at a speed of 15 mm/min until reaching 10 N and then continued at a speed of 5 mm/min. The μTBS data, expressed in megapascals (MPa), were recorded by dividing the applied force at the time of rupture (peak load) by the bond area (mm^2^). Specimens that broke during transportation, handling, and/or assembly were excluded from the study.

After the test, the fractured samples were examined under an optical microscope (HMV-2, Shimadzu, Kyoto, Japan) at a magnification of 200× by the same evaluator, and the fracture modes were classified as follows: A: Adhesive failure, which is a fracture within the adhesive layer. C: Cohesive failure, either within the resin composite or dentin. M: Mixed failure, where the fracture involves more than one material. P: Pretest failure, indicating that the fracture occurred before testing the sample. G: Failure outside the testing area, where the sample was attached to the device [30].

### 2.5. Data Analysis

The data were analyzed for normality using the Shapiro–Wilk test. Since a normal distribution was not found in all groups, the nonparametric Kruskal–Wallis test and the Wilcoxon signed-rank test were used. A significance level of 5% (*p* < 0.05) was adopted for the analysis, and STATA version 14 was utilized. The μTBS data were analyzed using a descriptive approach.

## 3. Results

### 3.1. Surface Roughness

After the application of statistical tests, it was found that there were no statistically significant differences between the groups regarding initial Ra, as they maintained results between 0.1 and 0.13 µm. However, statistically significant differences were observed in relation to the final results, indicating that the Constic resin presented the highest superficial Ra. Differences were also observed in the initial and final results (intragroup) for all the tested resins (Table 1), with an increase in Ra in all the groups.

### 3.2. Color Assessment

The color parameters of the groups are shown in Table 2 and Table 3. In the comparison according to the CIELAB formula (*p* = 0.080) and CIEDE200 (*p* = 0.144), there was no statistically significant difference (*p* > 0.05) between the groups (G1A, G2A, and G3A). The color variation values were analyzed on the basis of 50%:50% perceptibility and 50%:50% acceptability criteria, with thresholds set at 1.2 and 2.7 for ΔEab and 0.8 and 1.8 for ΔE 00. All the results fell within the acceptable limits, as shown in Figure 1 and Figure 2.

### 3.3. Microtensile Test

A significant difficulty was observed in obtaining the required number of sticks for the microtensile test in the groups where the SARF restorations were performed. During the cutting process of the restored dental element, these materials exhibited fragility, resulting in detachment of the dentin in the first longitudinal cut or during the formation of the stick in the second cut. As a result, only 8 sticks were obtained in the Yflow SA resin group, and 11 sticks were obtained in the Constic resin group. This fragility indicates early failure in the restorations prior to the microtensile test. In contrast, obtaining the necessary sample size of sticks for the CFR was much easier, with no fractures occurring during cutting, resulting in a sample size of 35 sticks.

This fact complicated the statistical analysis of these data, which were then discussed through a descriptive comparison of the mean μTBS values. The analysis revealed that TNF had the highest mean μTBS (33.669 ± 6.453 MPa), followed by Constic (10.649 ± 4.406 MPa) and Yflow SA (8.170 ± 3.781 MPa) (Figure 3). The TNF group has a larger sample size (n = 35), which increases the reliability and precision of the mean estimates. This suggests that the mean μTBS of this group is more representative of the materials studied. The fracture patterns are shown in Figure 4.

## 4. Discussion

This in vitro study evaluated the behavior of SARF in terms of Ra, color stability after simulated brushing, and μTBS to dentin. The properties of SARFs are crucial to the long-term success of dental restorations, particularly the properties that were evaluated in this study. The surface topography and color stability are critical factors for the clinical success of restorative dental treatments. An ideal restoration should have a smooth surface to prevent biofilm adhesion and maintain color stability over time [19,31]. Brushing is a common practice that significantly impacts the surface properties of restorative materials, affecting both their appearance and Ra [19]. Despite the time-saving advantage of eliminating the acid etching and adhesive system application steps, it is necessary to study the bond strength of SARFs to dental substrates, a characteristic that has been shown to be weak and may compromise the longevity of restorative procedures [22]. The findings of this study not only add to the body of knowledge but also provide practical insights that can guide clinicians in making more informed choices regarding material selection and patient care.

The initial hypothesis that there would be no significant difference in Ra among the tested resin materials after simulated brushing was refuted. Specifically, the Constic resin demonstrated a notably greater Ra following the abrasive challenge, showing a significant increase in Ra (0.7 µm) compared with both the TNF (0.31 µm) and Yflow SA (0.18 µm) resins. This observed discrepancy can be attributed to variations in the composition of the organic matrix and the relatively low filler content present in the materials [18,32]. Past research has underscored the impact of organic matrix composition on material properties, particularly in relation to abrasion resistance. Notably, higher levels of Bis-GMA may contribute to a reduction in abrasion resistance, whereas increased amounts of UDMA and TEGDMA could increase resistance to abrasion [18]. This suggests that differences in the resin composition among the tested materials likely played a pivotal role in the observed variations in Ra following simulated brushing.

Lai et al. [19] reported a significant increase in Ra values in the SAFR group (Vertise Flow) compared with two nanohybrid resins (GrandioSO Flow and G-aenial Universal Flow) after simulated brushing, with Ra values of 0.21 µm, 0.17 µm, and 0.11 µm, respectively. In our study, there were two resins in the SARF group: one (Constic), which had a greater roughness than the CFR (TNF), and another (YFlow SA), which had a lower roughness, reinforcing that differences in composition can influence the behavior of SARFs after the abrasive challenge. Similar results were reported by Malavasi et al. [18], who compared self-adhesive flowable resins (vertise flow and fusio liquid dentin) with a nanoparticulate resin (Filtek 350) after 20,000 cycles of simulated brushing. A greater increase in Ra was observed in the SAFR group, with values of 0.24 µm, 0.15 µm, and 0.13 µm, respectively. Notably, Costa et al. [33] emphasized that Ra exceeding 0.2 µm could heighten susceptibility to biofilm accumulation and bacterial adhesion, potentially leading to the development of secondary caries and periodontal inflammation. However, they reported that Ra alterations ranging between 0.22 and 0.24 µm are considered clinically acceptable [33]. In this context, the Ra values of the Yflow SA self-adhesive resins were the only ones that met clinically acceptable standards.

The second null hypothesis, which posited no differences in color among the studied composite resins after simulated brushing, was accepted when the CIELAB (ΔEab) and CIEDE2000 (ΔE00) tools were applied (Table 2 and Table 3). According to Table 2, the mean ΔEab values for the resins after simulated brushing were as follows: TNF (1.977), Constic (2.307), and Yflow SA (1.798). Statistical analysis revealed that the differences between these groups were not significant (*p* = 0.080). According to Table 3, the mean ΔE00 values for the resins tested after the abrasivity challenge were 0.962 for TNF, 1.140 for Constic, and 1.058 for Yflow SA, and the differences among these groups also failed to reach statistical significance (*p* = 0.144). The CIEDE 2000 method yielded lower mean ΔE values and offered a more consistent assessment of color stability, reflecting its superior accuracy in aligning with human visual perception. These findings align with those of Malavasi et al. [18], where self-adhesive flowable resins (SAFRs) (Vertise Flow and Fusio Liquid Dentin) showed no statistically significant changes compared with a nanoparticulate resin (Filtek 350) in terms of color variation (ΔEab). Similarly, the study by Sanal and Kilinc [34] comparing SAFR (Constic, Vertise Flow, and Fusio Liquid Dentin) with conventional resin (Filtek Supreme XTE) in ceramic repairs corroborates these results. In contrast, in the study by Lai et al. [19], the evaluated SARF (vertise flow) showed greater color change (ΔEab) after the simulated abrasive challenge. Importantly, in laboratory studies, intrinsic patient factors such as diet are not considered. The minor observed alterations, although statistically nonsignificant, may be attributed to the optical properties resulting from internal reactions of the restorative material [18] and simulated brushing.

The literature contains studies in which self-adhesive flowable resin (fusion liquid dentin) has been shown to demonstrate better color stability clinically for more than one year than conventional resin (TNF). In the TNF group, slight color mismatch, hue, or translucency was noted in 5 out of 30 restorations, albeit within the normal range of adjacent tooth structure [9]. Arregui et al. [35], on the other hand, examined two SARFs (Vertise Flow and experimental SAFRs) and one nanohybrid resin (premise flow) and noted statistically significant differences in color variation (ΔE00) between the experimental SAFR group (12.25) and the Vertise Flow (6.57) and Premise Flow (3.47) groups. These variations could be attributed to the different methodologies employed, including immersion in high-temperature water for 30 days.

In this study, we assessed the total color difference (ΔE) using both the CIEDE 2000 (ΔE00) and CIELAB (ΔEab) formulas, considering acceptability (AT) and perceptibility (PT) (Figure 1 and Figure 2). A ΔEab value below 1.2 indicated that 50% of the observers could detect differences, whereas the other 50% could not [25]. For acceptability, a shade difference under 2.7 was deemed clinically acceptable [25]. The CIEDE 2000 formula thresholds for perceptibility and acceptability were set at 0.8 and 1.8, respectively [25]. Several studies have reported better agreement between the CIEDE2000 formula and visual findings (95% agreement) than between the CIEDE2000 formula and the CIELAB formula (75%), supporting its use in tooth color research [25]. Our results revealed ΔE values below both the PT and AT thresholds for both formulas. Importantly, the standard deviation found between the samples was lower when the CIEDE2000 system was used, probably because this method has lower perceptibility and acceptability limits than CIELAB does, indicating that it is capable of better distinguishing color differences. Therefore, on the basis of these interpretations, the results suggest that the color differences among the groups are clinically acceptable and do not pose a significant issue in terms of aesthetics for the studied composite resin restorations.

Upon interpreting our results, we observed the rejection of the third null hypothesis of this study, which suggested that self-adhesive resins would exhibit μTBSs comparable to the CFRs established on the market. This rejection was evidenced by two main factors: the occurrence of early restoration failure and the lower average μTBS in the groups restored with the SARF. The occurrence of restoration failure during the pre-experimental period was reported in a previous study [13,16]. The fusio liquid dentin (FLD) group was the only group that did not exhibit failure, whereas the other self-adhesive resins had failure frequencies of 3 out of 10 restorations and 5 out of 10 restorations, respectively [13]. The presence of this phenomenon may bias the results, leading to a possible overestimation of the values. Consequently, the authors chose to consider the μTBS value to be 0 MPa in cases of failure [13,16]. In the present study, we decided to exclude samples that fractured before the microtensile test from the analyses. This procedure was adopted to ensure the integrity and reliability of the results. The inferior performance of self-adhesive resins compared with CFR in terms of μTBS is supported by several previous studies [13,15,16,27,36].

Cengiz and Unal [37] conducted a microtensile μTBS test using Vertise Flow (VF) and Fusion Liquid Dentin (FLD) self-adhesive resins with and without the use of self-etch or total-etch dentin adhesives. The groups that did not receive dentin adhesive, as recommended by the manufacturers, had lower μTBS values. Notably, the use of a universal adhesive with the total-etch technique improved the μTBS in all the evaluated groups. This study used a sample size of n = 10 per group, similar to the values found in the Constic (n = 11) and Yflow SA (n = 8) groups in our research. The mean μTBS found without the use of the adhesive system was 8.06 for VF and 5.38 for FLD, values that are close to those reported in our study: 11.027 for Constic and 8.17 for Yflow SA.

Peterson et al. [36] reported that the adhesive tags in self-adhesive resins were thin and spaced, in contrast to those in conventional resins, where they were partially branched. This study highlighted the poor interaction with dental tissue in the presence of a smear layer, raising doubts about the viability of self-adhesive resins as direct restorative materials [36,38]. Importantly, these findings were revealed in a study that used flat dentin; under conditions closer to clinical reality, the failure rates may be even greater [39]. However, when applied to a dental surface without a smear layer, self-adhesive resin cements (with a mechanism similar to that of SARF) produce longer and more uniform resin tags [39]. This behavior raises doubts about the similarity of these materials to self-etch adhesives, as suggested by manufacturers [12]. A possible explanation for this recurring result is the high viscosity of self-adhesive resins, which may hinder deep penetration into dentinal tubules and between collagen fibers [13,40]. A systematic review addressing the μTBS of self-adhesive resins revealed that this inferior performance could be improved in dental enamel with the use of total acid conditioning [41].

This study, while informative, has some limitations worth noting. As a laboratory-based assay, it may not fully replicate the complex conditions encountered in the oral cavity, particularly those associated with more intricate cavities beyond the scope of our simulated flat dentin model. The toothpaste used in this in vitro study was diluted in distilled water, which is quite different from the oral cavity, where saliva contains enzymes, ions, and proteins. Additionally, the sample size in the SARF groups for the μTBS test could impact the generalizability of our findings to clinical settings. However, despite these limitations, our findings offer valuable insights. These findings provide a foundation for future in vitro investigations aimed at confirming the clinical safety and efficacy of SARF materials. By identifying potential areas of improvement and highlighting areas for further research, our study contributes to advancing the understanding and application of the SARF in clinical dentistry.

## 5. Conclusions

This in vitro study demonstrated that SARFs show promise in terms of color stability, as their performance was comparable to that of the CFR control after simulated brushing. However, regarding the Ra values, one SARF (Constic) exhibited greater roughness, whereas the other SARF (Yflow SA) presented lower roughness than the CFR. This suggests that future research should explore differences in manufacturers’ compositions or application techniques to enhance SARF performance. The μTBS tests revealed that the bonding strength of the SARFs was weaker than that of the CFR when an adhesive system was used. Specifically, the TNF demonstrated the highest μTBS, whereas the Constic and Yflow SA SARFs presented considerably lower values. These lower μTBS values for SARFs highlight a potential limitation for their clinical use, particularly in scenarios requiring strong dentin adhesion. One limitation of this study was the difficulty in obtaining sticks for the microtensile bond strength test due to the fragility of the SARFs, which may affect the robustness of the data. Additionally, while the simulated brushing protocol provides valuable insights, it may not fully replicate the clinical complexity of restorative material wear in the oral environment.

## Figures and Tables

**Figure 1 polymers-16-02556-f001:**
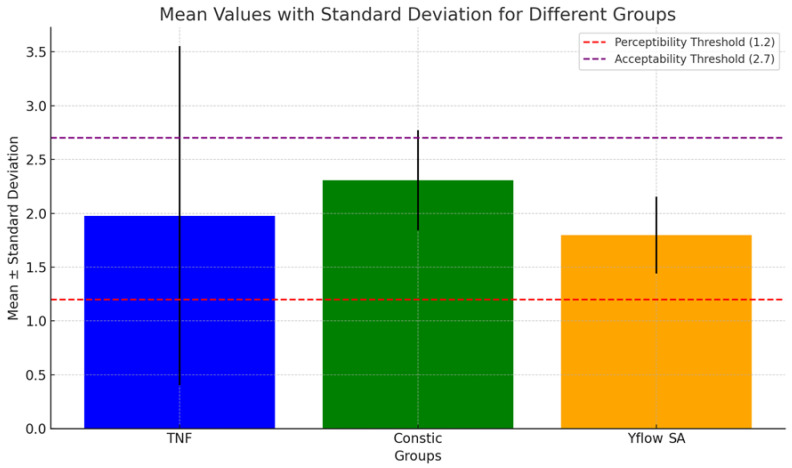
Bar chart showing the mean values with standard deviations for different groups (TNF, Constic, and Yflow SA) analyzed for color stability using the CIELAB system (ΔEab). The red dashed line represents the perceptibility threshold (ΔEab = 1.2), and the purple dashed line represents the acceptability threshold (ΔEab = 2.7). The black lines indicate the standard deviation, showing the variation around the mean for each group.

**Figure 2 polymers-16-02556-f002:**
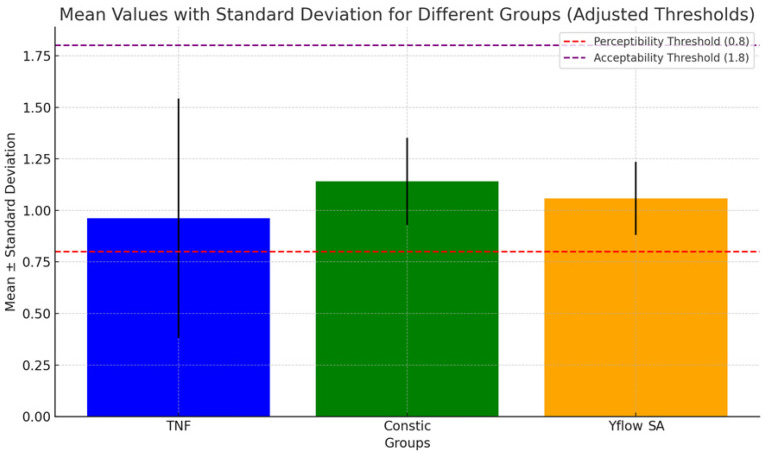
Bar chart showing the mean values with standard deviations for different groups (TNF, Constic, and Yflow SA) analyzed for color stability using the CIEDE 2000 system (ΔE00). The red dashed line represents the perceptibility threshold (ΔEab = 0.8), and the purple dashed line represents the acceptability threshold (ΔEab = 1.8). The black lines indicate the standard deviation, showing the variation around the mean for each group.

**Figure 3 polymers-16-02556-f003:**
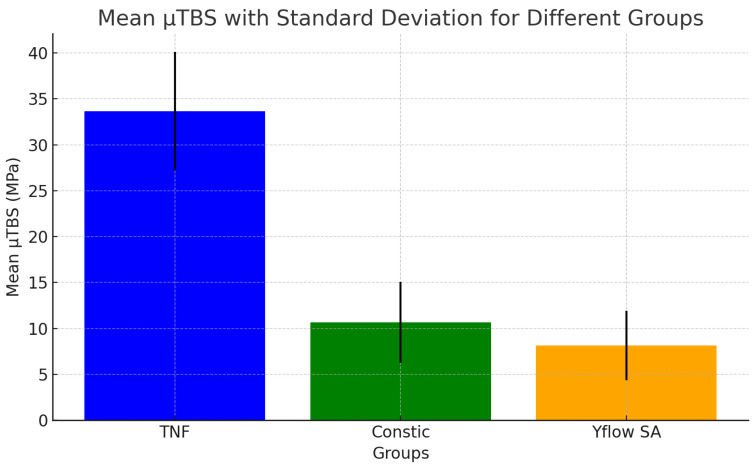
Bar chart showing the mean μTBS values (MPa) and standard deviations of the groups (TNF, Constic, and Yflow SA).

**Figure 4 polymers-16-02556-f004:**
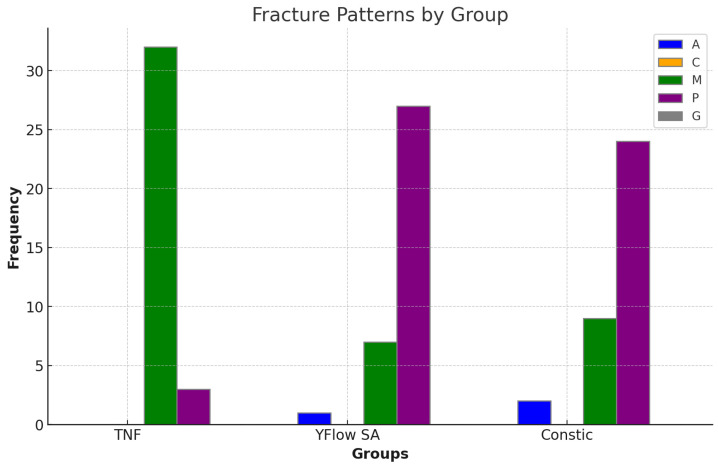
Bar chart showing the distribution of fracture patterns of the groups (TNF, Yflow SA, and Constic) after the dentin bond strength test. Legend: A: Adhesive failure, which is a fracture within the adhesive layer. C: Cohesive failure, either within the resin composite or dentin. M: Mixed failure, where the fracture involves more than one material. P: Pretest failure, indicating that the fracture occurred before testing the sample. G: Failure outside the testing area, where the sample was attached to the device.

**Table 1 polymers-16-02556-t001:** Means and standard deviations of surface roughness (µm) before and after simulated toothbrushing.

Groups	Before	After	*p*
TNF	0.101 ± 0.019	0.316 ± 0.070	0.005 *
Constic	0.111 ± 0.019	0.702 ± 0.282	0.005 *
Yflow AS	0.131 ± 0.053	0.184 ± 0.065	
*P*	0.051	<0.001 *	0.012 *

* Indicates a statistically significant difference (*p* < 0.05).

**Table 2 polymers-16-02556-t002:** ΔEab–CIELAB—Comparison of initial and after simulated toothbrushing color difference measures between groups.

Groups	Mean ± dp
TNF	1.977 ± 1.575
Constic	2.307 ± 0.464
Yflow AS	1.798 ± 0.356
*P*	0.080

**Table 3 polymers-16-02556-t003:** ΔE 00–CIEDE 2000—Comparison of initial and after simulated toothbrushing color difference measures between groups.

Groups	Mean ± dp
TNF	0.962 ± 0.580
Constic	1.140 ± 0.213
Yflow AS	1.058 ± 0.177
*P*	0.144

## Data Availability

All the data are available within the manuscript.

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
