# Peer review of "Assessment of Surface Roughness, Color, and Bonding Efficacy: Self-Adhesive vs. Conventional Flowable Resin"

_polymers, 2024, doi:10.3390/polym16182556_

Round 1

Reviewer 1 Report

Comments and Suggestions for Authors

This in vitro study aimed to analyze the surface roughness (Ra) and color stability (ΔEab, ΔE00) following simulated mechanical brushing, and to evaluate the microtensile (μTBS) to dentin of Self Adhering Resin Flowable (SARF). The selected materials were Constic, Yflow AS, and Tetric N flow (TNF/control). Thirty composite resin cylinders were fabricated for surface property evaluation. I believe this paper can be published in the journal if the authors address the comments effectively and revise their manuscript. 

1- The Introduction Section in its current form is not adequate. In this context, I strongly recommend the author(s) to analyze and discuss separately each cited paper. Besides, the differences/advantages of the present investigation compared to other literature works should be written out at the end of this Section in a much more detailed and comprehensive manner.

2- English of the manuscript needs polishing.

3- The Conclusions Section performs the findings of this work in a rather brief manner. It should become more thorough and explanatory. Further, I invite the author(s) to add a paragraph on the motives and prospects that this work provides for future research.

4- Please add more discusion about Tables 2 and 3. 

5- Limitations and future work of the study needs to be included in the conclusion section. 

Comments on the Quality of English Language

English of the manuscript needs polishing.

Author Response

Reviewer(s)' Comments to Author:

Reviewer 1

This in vitro study aimed to analyze the surface roughness (Ra) and color stability (ΔEab, ΔE00) following simulated mechanical brushing, and to evaluate the microtensile (μTBS) to dentin of Self Adhering Resin Flowable (SARF). The selected materials were Constic, Yflow AS, and Tetric N flow (TNF/control). Thirty composite resin cylinders were fabricated for surface property evaluation. I believe this paper can be published in the journal if the authors address the comments effectively and revise their manuscript. 

Comments 1- The Introduction Section in its current form is not adequate. In this context, I strongly recommend the author(s) to analyze and discuss separately each cited paper. Besides, the differences/advantages of the present investigation compared to other literature works should be written out at the end of this Section in a much more detailed and comprehensive manner.

Answer: Thanks for the suggestion. The articles used in the introduction were the best explored and discussed. Furthermore, a paragraph highlighting the innovations of this study was added at the end of this section.

“Compared with traditional bonding techniques, the microtensile μTBS (μTBS) of SAFR remains a subject of investigation, with conflicting findings [13-15]. Brueckner et al. [13] conducted a dentin and enamel bonding strength test after simulated aging with thermocycling, comparing SAFR with a conventional flowable resin used with an adhesive system. The SAFRs evaluated (vertise flow, fusion liquid dentin, and an experimental SAFR) exhibited lower adhesion performance to dental tissues than did conventional flowable resins in Class V cavities. In turn, Abdelraouf, Mohammed, and Abdelgawad [14], in a similar study in which the testing area was prepared after the vestibular face of the included molars was cut, also reported that the evaluated SAFR (Dyad-flow) had inferior bonding strength to dental substrates compared with conventional flowable resins. Hayashi et al. [10] reported that the adhesion values of SARFs are greater when prior acid etching is performed, according to an in vitro study conducted on bovine teeth. This study highlights the need for further investigations into specific resin formulations and insertion techniques to assess their clinical viability. For example, it was found that inserting one SARF (fusion liquid dentin) was easier than inserting another (vertise flow) [16]. Elraggal et al. [17] conducted shear and microtensile bond strength tests to evaluate the adhesion of SARFs to dentin after thermocycling. The resins tested (Surefil One and Vertise Flow) demonstrated inferior results compared to conventional flowable resin. Studies evaluating the μTBS of restorative materials are crucial for subsequent clinical research, as stated by Brueckner et al. [13] and Elraggal et al. [17]. In vitro studies reflect the clinical performance of resins, and we can only use them in vivo after promising laboratory results are obtained”.

“… SARFs demonstrated high luster and a smooth surface after polishing [20]. However, after 60 minutes of simulated brushing with an electric toothbrush and Colgate Total toothpaste, these characteristics were altered. The evaluated SARF exhibited an Ra value above 0.2 µm, leading to surface irregularities, filler dislodgement, and protrusion [20]. Malavasi et al. [19] conducted a simulated brushing, where specimens were brushed for 20,000 cycles using a soft-bristled toothbrush and Colgate Total toothpaste. After the abrasive challenge, SARFs exhibited higher (0,24 and 0.15 µm) surface roughness values compared to the conventional resin (0.13 µm) used as a control. This increase in roughness was attributed to the smaller filler particles in the conventional resin, which provided better support and protection for the fillers. As the abrasive process removed the organic matrix of the resins, the larger filler particles in the SARFs were more exposed, leading to greater surface irregularities [19]. Increased roughness of restorative materials makes the surface more susceptible to plaque accumulation, staining, and deterioration of the material's aesthetic properties [21]. Color represents an important aspect of the aesthetic success of restorations. A previous study [20] highlighted that simulated brushing affected the color stability of self-adhesive flowable resins (SARFs), with the SARFs showing the highest ΔE values after the abrasive challenge. This rise in ΔE was attributed to the abrasive process itself; during brushing, the specimens were immersed in a slurry (toothpaste and distilled water), which can lead to water absorption and potentially impact color stability [20].”

“…can provide essential information for further testing and ensure safety in future clinical trials [23]. This study provides insights into the performance of SARFs by comprehensively evaluating their surface roughness (Ra), color stability, and microtensile bond strength (μTBS) in comparison to CFRs. Through these evaluations, the results will either reaffirm or challenge the use of self-adhesive resins, offering a clearer understanding of their suitability and effectiveness in clinical practice. While previous research has focused on specific aspects of SARFs, such as bond strength or color stability, this study adopts a more holistic approach by simultaneously assessing these critical factors. The use of both the CIELAB and CIEDE2000 methods for color assessment contributes to a more nuanced understanding of color stability across various SARFs and CFRs. Furthermore, the study addresses the mechanical performance of SARFs in terms of μTBS, highlighting a gap in the literature, as no studies evaluating the μTBS of Yflow SA (a specific SARF) were found. The findings of this study can provide scientific evidence on the performance of SARFs, contributing to…”

Comments 2- English of the manuscript needs polishing.

Answer: Thank you for your suggestion, the English has been revised.

Comments 3- The Conclusions Section performs the findings of this work in a rather brief manner. It should become more thorough and explanatory. Further, I invite the author(s) to add a paragraph on the motives and prospects that this work provides for future research.

Answer: Thanks for the suggestion, the conclusion has been updated.

“This in vitro study demonstrated that SARFs show promise in terms of color stability, as their performance was comparable to that of the CFR control after simulated brushing. However, regarding Ra values, one SARF (Constic) exhibited higher roughness, while the other SARF (Yflow SA) showed lower roughness compared to the CFR. This suggests that future research should explore differences in manufacturers' compositions or application techniques to enhance SARF performance. The μTBS tests revealed that SARFs had weaker bonding strength compared to CFRs utilizing an adhesive system. Specifically, TNF demonstrated the highest μTBS, while Constic and Yflow SA SARFs showed considerably lower values. These lower μTBS values for SARFs highlight a potential limitation for their clinical use, particularly in scenarios requiring strong dentin adhesion. One limitation of this study was the difficulty in obtaining sticks for the microtensile bond strength test due to the fragility of SARFs, which may affect the robustness of the data. Additionally, while the simulated brushing protocol provides valuable insights, it may not fully replicate the clinical complexity of restorative material wear in the oral environment”.

Comments 4- Please add more discussion about Tables 2 and 3. 

Answer: Thank you for the suggestion, we have improved the mention of tables in the discussion session.

“The second null hypothesis, which posited no differences in color among the studied composite resins after simulated brushing, was accepted when applying the CIELAB (ΔEab) and CIEDE2000 (ΔE00) tools (Tables 2 and 3). According to Table 2 the mean ΔEab  for the resins after simulated brushing were: TNF (1.977), Constic (2.307) and Yflow SA (1.798), the statistical analysis revealed that the differences between these groups were not significant (p = 0.080). And, according table 3 the mean ΔE00 for the tested resins after abrasive chalenge were: TNF (0.962), Constic (1.140) and Yflow SA (1.058), the differences among these groups also failed to achieve statistical significance (p = 0.144). The CIEDE 2000 method yielded lower mean ΔE values and offered a more consistent assessment of color stability, reflecting its superior accuracy in aligning with human visual perception. These findings align with those of Malavasi et al. [19], where self-adhesive flowable resins (SAFRs) (Vertise Flow and Fusio Liquid Dentin) showed no statistically significant changes”

Comments 5- Limitations and future work of the study needs to be included in the conclusion section.

Answer: Thanks for the suggestion, the conclusion has been updated.

“This in vitro study demonstrated that SARFs show promise in terms of color stability, as their performance was comparable to that of the CFR control after simulated brushing. However, regarding Ra values, one SARF (Constic) exhibited higher roughness, while the other SARF (Yflow SA) showed lower roughness compared to the CFR. This suggests that future research should explore differences in manufacturers' compositions or application techniques to enhance SARF performance. The μTBS tests revealed that SARFs had weaker bonding strength compared to CFRs utilizing an adhesive system. Specifically, TNF demonstrated the highest μTBS, while Constic and Yflow SA SARFs showed considerably lower values. These lower μTBS values for SARFs highlight a potential limitation for their clinical use, particularly in scenarios requiring strong dentin adhesion. One limitation of this study was the difficulty in obtaining sticks for the microtensile bond strength test due to the fragility of SARFs, which may affect the robustness of the data. Additionally, while the simulated brushing protocol provides valuable insights, it may not fully replicate the clinical complexity of restorative material wear in the oral environment”.

Reviewer 2 Report

Comments and Suggestions for Authors

1. In the abstract, please include the quantitative data for the lowest and highest groups to enhance understanding.
2. Add a keyword related to color in the keywords section.
3. It would be better to transfer the data from Table 1 into the text. I recommend making this change.
4. The results section should be divided into different parts for better presentation.
5. All figures and plots should be re-drawn to be more interactive. Although the data appears to be good, the presentation is weak.
6. The conclusion section is completely inappropriate and needs to be re-written. It is currently very weak.

Based on the above comments, I cannot accept the manuscript in current form.

Good luck.

Author Response

Reviewer 2

Comments 1- In the abstract, please include the quantitative data for the lowest and highest groups to enhance understanding.

Answer: Thank you for your suggestion. The lowest and highest roughness data of the evaluated materials have been included in the abstract as recommended.

“Statistically significant differences in Ra among the groups, with Constic exhibiting the highest Ra value (0,702 µm; p < 0.05), while Yflow AS showed the lowest Ra value (0,184 µm).”

Comments 2- Add a keyword related to color in the keywords section.

Answer: Thank you for your suggestion. The keyword "color" has been added to the keywords section.

“Keywords: Self adhesive flowable composite; Toothbrushing; Microtensile bond Strength; Surface Properties; Color.”

Comments 3- It would be better to transfer the data from Table 1 into the text. I recommend making this change.

Answer:  Thank you for the suggestion. The information contained in Table 1 has been transformed into text as recommended and incorporated into the article as follows:

Subsection: Materials and methods

“…Constic (DMG, Hambug, Germany; Lot No. 232801) and Yflow SA (Yller Biomateriais, Pelotas, RS, Brazil; Lot. No. 00009968) in comparison with the CFR Tetric N-flow (TNF) (Ivoclar Vivadent, Barueri, SP, Brazil; Lot No. Z020DP, Z019WR). The Constic consists of barium glass in a matrix of dental resins based on Bis-GMA, EBADMA, UDMA, HEMA, TEGDMA, HDMA, and MDP, along with pigments, additives, and a catalyst (DMG). The Yflow contains methacrylate monomers, acid monomers, inorganic fillers, pigments, initiators, and stabilizers (Yller biomaterials) and TNF contains dimethacrylates (including TEGDMA), barium oxide, ytterbium trifluoride, highly dispersed silica, mixed oxides, pigments, catalysts, and stabilizers (Ivoclar Vivadent). The...”

Subsection: Simulated toothbrushing

“…cycle. The toothpaste used was Colgate Cavity Protection (Colgate-Palmolive Company, New York, USA), which includes 1500 ppm of fluoride, calcium carbonate, sodium lauryl sulfate, sodium saccharin, tetrasodium pyrophosphate, sodium silicate, polyethylene glycol, sorbitol, carboxymethyl cellulose, methylparaben, propylparaben, aromatic composition, and water. It also contains sodium monofluorophosphate (MFP®). The toothpaste was applied as a suspension and diluted in distilled water at a ratio of 2:1 using a mechanical agitator. The...”

Subsection: Microtensile test:

“…a Demetron radiometer (Model 100, Demetron, USA), resulting in the formation of a block approximately 4.0 mm in height. The restorative technique for each composite resin was performed according to the manufacturer's instructions. For TNF, the procedure involves applying adhesive to the enamel and dentin, then placing the resin into the cavity using the incremental technique. Each increment should be between 1.5 and 2 mm thick, with a curing time of 20 seconds per increment using a halogen light with a minimum intensity of 500 mW/cm². For Constic, the material is applied to the cavity surface by pressing the syringe and rubbing the resin for 25 seconds to form a thin layer (0.5 mm). Any excess is removed if necessary, followed by light-curing for 20 seconds. Additional increments of up to 2 mm are then applied and cured for 20 seconds each. For Yflow, after dental cleaning, the product is directly applied into the cavity using a syringe with an applicator tip. The resin is placed in increments of up to 2 mm, with each increment being light-cured for 40 seconds.”

Comments 4- The results section should be divided into different parts for better presentation.

Answer: Thanks for the suggestion, the following subsections have been added to the results section:3.1 Surface roughness; 3.2 Color assessment; 3.3 Microtensile test.

Comments 5- All figures and plots should be re-drawn to be more interactive. Although the data appears to be good, the presentation is weak.

Answer: Thank you for the suggestion, all the figures have been updated in more interactive charts.

Comments 6- The conclusion section is completely inappropriate and needs to be re-written. It is currently very weak.

Thanks for the suggestion, the conclusion has been updated.

“This in vitro study demonstrated that SARFs show promise in terms of color stability, as their performance was comparable to that of the CFR control after simulated brushing. However, regarding Ra values, one SARF (Constic) exhibited higher roughness, while the other SARF (Yflow SA) showed lower roughness compared to the CFR. This suggests that future research should explore differences in manufacturers' compositions or application techniques to enhance SARF performance. The μTBS tests revealed that SARFs had weaker bonding strength compared to CFRs utilizing an adhesive system. Specifically, TNF demonstrated the highest μTBS, while Constic and Yflow SA SARFs showed considerably lower values. These lower μTBS values for SARFs highlight a potential limitation for their clinical use, particularly in scenarios requiring strong dentin adhesion. One limitation of this study was the difficulty in obtaining sticks for the microtensile bond strength test due to the fragility of SARFs, which may affect the robustness of the data. Additionally, while the simulated brushing protocol provides valuable insights, it may not fully replicate the clinical complexity of restorative material wear in the oral environment”.

Reviewer 3 Report

Comments and Suggestions for Authors

The current work is somewhat at the edge between a scientific paper and a technical report. It does not contain major flaws but I suggest to:

1.      Start of Results and Discussion: make a link to the state of the art. Explain the relevance of the analysis for a general reader.

2.      The captions need to be much more detailed.

3.      The first figures need more discussion, specifically vs. target values.

4.      The conclusions are too reporting style

Author Response

Reviewer 3

The current work is somewhat at the edge between a scientific paper and a technical report. It does not contain major flaws but I suggest to:

Comments 1-    Start of Results and Discussion: make a link to the state of the art. Explain the relevance of the analysis for a general reader.

Answer: Thanks for the suggestions. Changes have been made.

“This in vitro study evaluated behavior of SARF in terms of Ra, color stability after simulated brushing, and μTBS to dentin. The properties of SARF are crucial to the long-term success of dental restorations, particularly the properties that were evaluated in this study. The Surface topography and color stability are critical factors for the clinical success of restorative dental treatments. An ideal restoration should have a smooth surface to prevent biofilm adhesion and maintain color stability over time [20, 33]. Brushing is a common practice that significantly impacts the surface properties of restorative materials, affecting both their appearance and Ra [20]. Despite the time-saving advantage of eliminating the acid etching and adhesive system application steps, it is necessary to study the bond strength of SARFs to dental substrates, a characteristic that has been shown to be weak and may compromise the longevity of restorative procedures [22]. The findings of this study not only add to the body of knowledge but also provide practical insights that can guide clinicians in making more informed choices regarding material selection and patient care”.

Comments 2- The captions need to be much more detailed.

Answer: Thank you for the suggestion. All captions have been uptaded.

Table 1. Mean and standard deviation of surface roughness (µm) before and after simulated toothbrushing

Table 2. – ΔEab - CIELAB – Comparison of initial and after simulated toothbrushing color difference measures between groups.

 Table 3. - ΔE 00- CIEDE 2000- Comparison of initial and after simulated toothbrushing color difference measures between groups.

Figure 1. Bar chart showing mean values with standard deviation for different groups (TNF, Constic, and Yflow SA) analyzed for color stability using the CIELAB system (ΔEab). The red dashed line represents the perceptibility threshold (ΔEab = 1.2), and the purple dashed line represents the acceptability threshold (ΔEab = 2.7).

Figure 2. Bar chart showing mean values with standard deviation for different groups (TNF, Constic, and Yflow SA) analyzed for color stability using the CIEDE 2000 system (ΔE00). The red dashed line represents the perceptibility threshold (ΔEab = 0.8), and the purple dashed line represents the acceptability threshold (ΔEab = 1.8).

Figure 3. Bar chart showing mean μTBS values (Mpa) and standard deviations of groups (TNF, Constic and Yflow SA).

Figure 4.  Bar chart showing the distribution of fracture patterns of groups  (TNF, Yflow SA and Constic) after dentin bond strenght test.

Comments 3- The first figures need more discussion, specifically vs. target values.

Answer: Thank you for the suggestion, we added in the paragraph that explains figures 1 and 2 in the discussion a reflection on the standard deviations in the different color analysis systems used (CIELAB and CIEDE2000).

“...Our results showed ΔE values below both the PT and AT thresholds for both formulas. It is important to highlight that the standard deviation found between the specimens was lower when using the CIEDE2000 System, probably because this method has lower perceptibility and acceptability limits than CIELAB, indicating that it is capable of better distinguishing color differences. Therefore…”

Comments 4- The conclusions are too reporting style

Answer: Thanks for the suggestion, the conclusion has been updated.

“This in vitro study demonstrated that SARFs show promise in terms of color stability, as their performance was comparable to that of the CFR control after simulated brushing. However, regarding Ra values, one SARF (Constic) exhibited higher roughness, while the other SARF (Yflow SA) showed lower roughness compared to the CFR. This suggests that future research should explore differences in manufacturers' compositions or application techniques to enhance SARF performance. The μTBS tests revealed that SARFs had weaker bonding strength compared to CFRs utilizing an adhesive system. Specifically, TNF demonstrated the highest μTBS, while Constic and Yflow SA SARFs showed considerably lower values. These lower μTBS values for SARFs highlight a potential limitation for their clinical use, particularly in scenarios requiring strong dentin adhesion. One limitation of this study was the difficulty in obtaining sticks for the microtensile bond strength test due to the fragility of SARFs, which may affect the robustness of the data. Additionally, while the simulated brushing protocol provides valuable insights, it may not fully replicate the clinical complexity of restorative material wear in the oral environment.”

Reviewer 4 Report

Comments and Suggestions for Authors

11-       In the abstract section, it is better to mention only the main points.

22-      In the last paragraph of the introduction, please add a brief description of the results.

33-    Please elaborate results and discussion with reference to literature.

44-   It is better to compare your results with other reports in a table.

55-    The conclusion should also need to elaborate with more solid points related to outcomes of study.

66-     Fig.1: The error bar for TNF is too wide. What is your explanation, and how can this error be minimized? This observation is also noted in Fig. 2.

77-      The colors used in Fig. 4 are not clearly distinguishable. Please revise the graph to enhance comparability.

88-     What are your main novelties in comparison to Reference number 20?

Comments on the Quality of English Language

 Moderate editing of English language required.

Author Response

Reviewer 4

Comments 1-  In the abstract section, it is better to mention only the main points.

Answer: Thanks for the suggestion, the abstract has been updated to be more concise.

Abstract: This in vitro study aimed to analyze the surface roughness (Ra) and color stability (ΔEab, ΔE00) following simulated mechanical brushing and to evaluate the microtensile (μTBS) of self-adhering resin flowable (SARF) to dentin. The selected materials were Constic, Yflow AS, and Tetric N flow (TNF/control). Thirty composite resin cylinders were fabricated for surface property evaluation. Ra and color were assessed both before and after simulated brushing. The thresholds of 50:50% perceptibility and acceptability of color differences of the evaluated resins were assessed. For μTBS analysis, fifteen molars were selected, sectioned to expose flat dentin surfaces, and restored according to the manufacturers' instructions . The samples were subsequently sectioned into sticks for to microtensile testing. There were statistically significant differences in Ra among the groups, with Constic exhibiting the highest Ra value (0,702 µm; p < 0.05), whereas Yflow AS presented the lowest Ra value (0,184 µm). No statistically significant difference in color was observed among the groups (p > 0.05). The 50:50% perceptibility and acceptability thresholds were set at 1.2 and 2.7 for ΔEab and 0.8 and 1.8 for ΔE 00. All the results fell within the acceptable limits. The mean μTBS values of Constic, Yflow AS, and TNF were 10.649 MPa, 8.170 MPa, and 33.669 MPa, respectively. The study revealed increased Ra and comparable color stability among all the tested composite resins after abrasion. However, the SARF exhibited lower μTBS compared to conventional using an adhesive system.

Comments 2- In the last paragraph of the introduction, please add a brief description of the results.

Answer: Thanks for the suggestion. The introduction was revised.

“as they can provide essential information for further testing and ensure safety in future clinical trials [23]. This study provides insights into the performance of SARFs by comprehensively evaluating their surface roughness (Ra), color stability, and microtensile bond strength (μTBS) in comparison to CFRs. Through these evaluations, the results will either reaffirm or challenge the use of self-adhesive resins, offering a clearer understanding of their suitability and effectiveness in clinical practice. While previous research has focused on specific aspects of SARFs, such as bond strength or color stability, this study adopts a more holistic approach by simultaneously assessing these critical factors. The use of both the CIELAB and CIEDE2000 methods for color assessment contributes to a more nuanced understanding of color stability across various SARFs and CFRs. Furthermore, the study addresses the mechanical performance of SARFs in terms of μTBS, highlighting a gap in the literature, as no studies evaluating the μTBS of Yflow SA (a specific SARF) were found. The findings of this study can provide scientific evidence on the performance of SARFs, contributing to a better understanding of their characteristics and limitations. Thus...”

Comments 3-  Please elaborate results and discussion with reference to literature.

Answer: Thanks for the suggestion. The results and discussion section were revised to reference the literature more properly.   

Comments 4-  It is better to compare your results with other reports in a table.

Answer: Thanks for the interesting suggestion, but with the conclusion of this paper we feel that it may not be imperative to summarize results from similar studies in a table, since our results have already been discussed with other reports in the body of the discussion section.

Comments 5- The conclusion should also need to elaborate with more solid points related to outcomes of study.

Answer: Thanks for the suggestion, the conclusion has been updated.

“This in vitro study demonstrated that SARFs show promise in terms of color stability, as their performance was comparable to that of the CFR control after simulated brushing. However, regarding Ra values, one SARF (Constic) exhibited higher roughness, while the other SARF (Yflow SA) showed lower roughness compared to the CFR. This suggests that future research should explore differences in manufacturers' compositions or application techniques to enhance SARF performance. The μTBS tests revealed that SARFs had weaker bonding strength compared to CFRs utilizing an adhesive system. Specifically, TNF demonstrated the highest μTBS, while Constic and Yflow SA SARFs showed considerably lower values. These lower μTBS values for SARFs highlight a potential limitation for their clinical use, particularly in scenarios requiring strong dentin adhesion. One limitation of this study was the difficulty in obtaining sticks for the microtensile bond strength test due to the fragility of SARFs, which may affect the robustness of the data. Additionally, while the simulated brushing protocol provides valuable insights, it may not fully replicate the clinical complexity of restorative material wear in the oral environment”.

Comments 6- Fig.1: The error bar for TNF is too wide. What is your explanation, and how can this error be minimized? This observation is also noted in Fig. 2.

Answer: The higher standard deviation observed in Figure 1 is probably due to the use of the CIELAB system for color stability analysis. CIELAB does not perfectly align with how the human eye perceives color differences, particularly when it comes to small variations. In contrast, Figure 2 shows a reduced standard deviation, which can be attributed to the use of the CIEDE 2000 system. CIEDE 2000 incorporates adjustments such as weighting factors for lightness (L*), chroma (C*), and hue (H*), along with corrections for neutral colors and larger color differences. These refinements help minimize the errors that can occur with CIELAB, especially for certain colors, resulting in more consistent and reliable measurements. This note was added to the discussion.

“Our results showed ΔE values below both the PT and AT thresholds for both formulas. It is important to highlight that the standard deviation found between the specimens was lower when using the CIEDE2000 System, probably because this method has lower perceptibility and acceptability limits than CIELAB, indicating that it is capable of better distinguishing color differences. Therefore, based on these interpretations…”

Comments 7-  The colors used in Fig. 4 are not clearly distinguishable. Please revise the graph to enhance comparability.

Answer: Thanks for the suggestion, the figure has been updated.

Comments 8- What are your main novelties in comparison to Reference number 20?

Answer: Reference number 20 evaluated only one self-adhesive flowable resin (Kerr Vertise Flow) in comparison to conventional flowable resins, whereas our study evaluated two self-adhesive flowable resins (Constic and Yflow SA). In reference 20, regarding roughness before and after the abrasive challenge, the self-adhesive flowable resin was among those with the highest values. In contrast, in our study, the self-adhesive flowable resins exhibited similar roughness before the abrasive challenge. After the challenge, one self-adhesive flowable resin (Constic) showed higher roughness than the conventional resin, while another self-adhesive flowable resin (Yflow) showed lower roughness, raising the question that self-adhesive resins may behave differently depending on the manufacturer, which should be a topic for further research. Regarding color stability, the self-adhesive flowable resin showed the greatest color change after the abrasive challenge according to reference 20, whereas in our study, the evaluated resins did not show differences in color stability. Our study utilized both the CIELAB and CIEDE 2000 color systems for color stability analysis, while reference 20 only used CIELAB. Additionally, our study investigated bond strength to dentin, an important test for understanding the longevity of the restoration concerning adhesion to dental substrates. For better comprehension, the following information has been added to the paragraphs:

“Lai et al. [20], observed a significant increase in Ra values in the SAFR group (Vertise Flow) compared to two nanohybrid resins (GrandioSO Flow and G-aenial Universal Flow) after simulated brushing, with Ra values of 0.21 µm, 0.17 µm, and 0.11 µm, respectively. In our study, there were two resins in the SARF group: one (Constic) that showed higher roughness than the CFR (TNF), and another (YFlow SA) that showed a lower result, reinforcing that differences in composition can influence the behavior of SARFs after the abrasive challenge. Similar..”

“repairs corroborates these results. In contrast, in the study by Lai et al. [20], the evaluated SARF (Vertise Flow) showed greater color change after the simulated abrasive challenge. It is important…”

Round 2

Reviewer 2 Report

Comments and Suggestions for Authors

The revised manuscript can be published in the current form.

Reviewer 4 Report

Comments and Suggestions for Authors

 Accept in present form

Comments on the Quality of English Language

Minor editing of English language required.